# Quantification of the effects of long-term straw return on soil organic matter spatiotemporal variation: A case study in typical black soil region

Yang Yan[1], Wenjun Ji[1,2,3], Baoguo Li[1,2], Guiman Wang [4], Songchao Chen[5,6], Dehai Zhu[1,2], Zhong Liu[1]

[1]College of land Science and Technology, China Agricultural University, Beijing, 100193, China
[2]Key Laboratory of Agricultural Land Quality, Ministry of Natural Resources, Beijing, 100193, China
[3]State Key Laboratory of Remote Sensing Science, Aerospace Information Research Institute, Chinese Academy of Sciences, Beijing, 100101, China
[4]Lishu Metrological Bureau, Lishu, Jilin 158100, China
[5]InfoSol, INRAE, US 1106, Orléans F-4075, France
[6]ZJU-Hangzhou Global Scientific and Technological Innovation Center, Hangzhou 311200, China

*Correspondence to*: Wenjun Ji (wenjun.ji@cau.edu.cn)

**Abstract.** The straw return practice is essential to soil organic matter (SOM) accumulation in the black soil area with high carbon sequestration potential. However, due to lacking accurate spatial distribution of straw return, few studies have carried

out rigorous research on the impact of long-term straw return on SOM spatiotemporal variation on a regional scale. This study was carried out across an approximately 3000 km$^2$ area in Lishu County, Northeast China, a typical agricultural plain. Based on a total of 619 soil samples and 16 environmental covariates, the study mapped the spatial distributions of SOM in 2006 and 2018 by random forest (RF) and evaluated the effects of the interaction of soil properties, land use and straw return on SOM spatial-temporal variation. The results show that in the context of long-term straw return, the mean SOM content increased

from 18.93 g kg$^{-1}$ to 20.84 g kg$^{-1}$ during 2006–2018. And 74.49 % of the region had a significant increase (maximum: 24.41 g kg$^{-1}$) of SOM. The severest SOM loss occurred in the northwest due to the light texture and the transition from paddy fields to dryland. Nevertheless, for areas from paddy fields to dryland, the SOM loss decreased with the increased cumulative crop residue coverage. The SOM even increased by 1.79 g kg$^{-1}$ when the cumulative crop residue coverage reached 0.60–1.00. In addition, soil with higher initial SOM and sand content had a lower response to straw return. The study revealed that straw

return is beneficial to carbon sink in farmland and is a better way to prevent a carbon source caused by the conservation of paddy field to dryland.

## 1 Introduction

Soil organic matter (SOM) profoundly impacts carbon contents, cationic exchange capacity, water holding capacity, soil fertility, microorganisms, and soil structure (Ciais et al., 2011). Therefore, the SOM contents' spatial-temporal variation was

significant to global warming, soil quality, and ecosystem health (Ciais et al., 2011; Viscarra Rossel et al., 2014; Ondrasek et al., 2019), especially in black soil with rich SOM (Lugato et al., 2014; Amelung et al., 2020b). Recently, poor management

practices have resulted in SOM loss in the black soil area. Previous studies have reported that soil fertility decreased in black soil areas in North America and Eastern Europe (Russell et al., 2005; Fabrizzi et al., 2003). Meanwhile, a similar decline trend occurred in the black soil area in Northeast China (Wang et al., 2018). Therefore, rapidly and accurately quantifying the heterogeneity of SOM in the black soil region is necessary.

Conventional mapping involves laboriously constructing maps by planetable and alidade (Ahrens, 2008). It is time-consuming and laborious and thus cannot satisfy the growing demand for the latest soil spatial information. Jenny (1994) described the soil as follows: $soil\ property = f\ (climate, organism, relief, parent\ material, age)$ (Jenny, 1994). McBratney et al. (2003) proposed the $SCORPAN$ function model and described the soil as follows: $soil\ property = f(prior\ soil\ information, climate, organism, relief, parent\ material, age, location)$ (Mcbratney et al., 2003). Based on soil-forming theory, digital soil mapping (DSM) uses statistical and geospatial techniques to model the relationship between soil properties and environmental covariates at a high spatial resolution. By analyzing the relationships between soil properties and environmental factors, DSM models can be developed to predict the soil properties of areas where no soil data exist. Therefore, it offers a promising solution for predicting soil properties with high precision and tremendous speed (Hengl et al., 2015; Dou et al., 2019; Liang et al., 2019; Schulze and Schütte, 2020). Moreover, DSM can also incorporate the temporal component in soil property mapping by taking time as an index and comparing soil maps at two moments to identify changes in soil properties over time. This is particularly useful in understanding the impact of land use and management practices on soil properties and identifying areas where remediation may be necessary. Thus, the DSM method with environmental factors can accurately quantify the SOM spatial-temporal variation and measure the relationship between environmental covariates and SOM variation on a regional scale (Schillaci et al., 2017; Song et al., 2018; Zhou et al., 2019).

Nowadays, SOM variation under different land-use change and management practices has attracted increasing attention (Pan et al., 2010; Muñoz-Rojas et al., 2015). Some studies have explored reducing soil organic carbon (SOC) loss by straw return (West and Post, 2002; Liu et al., 2014; Wang et al., 2015; Amelung et al., 2020). Straw return is beneficial for retaining soil moisture and preventing soil wind erosion, especially in arid and semi-arid regions. In addition, the decomposition process of straw promotes the activity of microorganisms and is conducive to SOM accumulation (Chang et al., 2014; Lu et al., 2009; Wang et al., 2015). Conversely, previous scholars have reported that the influence of straw return on SOM accumulation is non-significant (Pittelkow et al., 2015; Poeplau et al., 2015; Powlson et al., 2011). This may be because adding organic matter to the soil has no effect on its chemical, chemical, and biological properties (Sosulski et al., 2011), or this practice may contribute to the SOM mineralization process and thus reduce SOM (Šimanský et al., 2019). The opposite result may be due to the various study areas with different soil properties, initial carbon content, land-use change, and straw return (Li et al., 2017; Berhane et al., 2020). In addition, these studies were mainly conducted on a field scale. On a regional scale, it is mostly through literature citation and policy enumeration to analyze the impact of straw return on SOM variation (Han et al., 2016; Zheng et al., 2015). However, few studies took the straw return as a variable to implement rigorous research on the effect of straw return on SOM variation due to lacking accurate spatial distribution of straw return.

In the study, the overall objective was to take a typical black soil area as a case to quantify the relationship between SOM accumulation and straw return on a regional scale. This study area has a long-term straw return background. Specific objectives included: a) evaluating the performance of random forest (RF) models with different groups of factors to develop the most robust model; b) analyzing the spatial-temporal variation of SOM during 2006–2018; and c) discussing the effects of long-term straw return on SOM variation under different soil types, soil texture, and land-use change.

## 2 Materials and methods


### 2.1 Study region

The study was conducted in Lishu County, Jilin Province, and its average elevation is 160 m (Fig. 1). The annual mean precipitation is 6.5 °C, the annual mean temperature is 553.5 mm, and the average yearly duration of sunshine is 2,541.4 hours. The region's climate is classified as semi-humid. In Lishu County, the soil parent material gradually changes from weathered

rocks and red sediments in the east to the loess-like sediment and loessal sub-sandy soil in the west, resulting in a regular distribution of soil type. Arenosols, Anthrosols, Phaezems, Luvisols, Cambisols, and Chernozems are the main soil types (World Reference Base for Soil Resources). Rainfall is the only source of water for crops growing in this region. The study area was located in the black soil region. However, the soil in this region was threatened by land degradation. In view of this, a research base was established in Lishu County, Jilin Province, China, in 2007, and the straw return technology was

popularized.

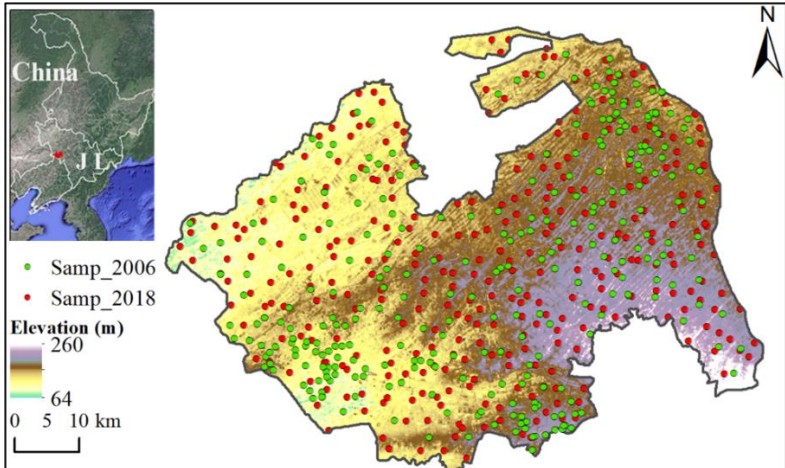

Figure 1: Schematic diagram of the geographical position of the study area and sampling sites. The background is from © Google Earth and the distribution of elevation was derived from the Resource and Environment Science and Data Center.

### 2.2 Soil data

The straw return measure has been implemented in Lishu County since 2007, so the SOM in 2006 and 2018 were selected to quantify the SOM change under the long-term straw return background. By taking into account the sample sites in the second

national soil survey, local landform, and soil types, a total of 366 sampling sites in 2006 were selected. Except for considering these factors, grid sampling was combined to select 355 sampling sites in 2018. These soil samples were collected on the surface (0–20 cm) from early October to mid-November in each year (from the harvest to the freezing). The corresponding longitude and latitude were also documented. The prediction error caused by the differences in sampling designs for the years 2006 and 2018 was not considered to make full use of legacy soil data (Ou et al., 2017; Sun et al., 2017; Nguemezi et al., 2021). Three kilograms of soil at each sampling site was air-dried. During the process of air drying, the soil samples were frequently turned over, and the intrusions outside the soil were removed. Sun exposure, acid, alkali, and dust pollution were strictly prohibited. After air drying and grinding, the soil samples were thoroughly mixed and passed through a 2 mm mesh. The samples were ground and sieved to separate a particle size fraction (0.25 mm) to determine SOC concentration with the external heating potassium dichromate volumetric method, which can then be multiplied by a conversion factor of 1.724 to obtain the SOM amount (Liu et al., 1996).

## 2.3 Environmental covariates

The study collected the available grids on the website as environmental factors. A 30 m resolution digital elevation model (DEM) was derived from the Resource and Environment Science and Data Center (http://www.resdc.cn/). Other terrain variables were calculated from DEM in the SAGA GIS software (Conrad et al., 2015), including terrain relief (TR), topographic wetness index (TWI), slope, aspect, profile curvature (PRC), multi-resolution valley bottom flatness (MrVBF) and plan curvature (PLC).

Landsat 5 TM and Landsat 8 TM were used to calculate the Normalized Difference Vegetation Index (NDVI) and Enhanced Vegetation Index (EVI) in 2006 and 2018. The average reflectance of three image bands (B1 (Blue), B3 (Red) and B4 (Near-infrared)) of Landsat 5 SR and Landsat 8 SR products spanning May to September were processed to calculate the NDVI by (B4 − B3) / (B4 + B3) and EVI by $2.5 \times (B4 − B3) / (B4 + 6 \times B3 − 7.5 \times B1 \times 1)$. The band calculation, and image clipping were conducted in Google Earth Engine (GEE), and the images with less than 6% of cloud coverages were selected.

Liu et al. (2021) provided a crop residue coverage map at a 10 m resolution in 2018 by combining radar indices and optical remote sensing indices. Crop residue cover refers to the ratio of the vertical projected area of crop residue in a field per unit area to the total surface area of this unit area, with value ranging from 0 to 1. Firstly, the study divided the study area into a sandy soil area and a clay soil area to reduce the influence of soil properties on radar echo and spectral reflectance. Six radar indices and five optical remote sensing indices were then calculated from a Sentinel-1 SAR image and a Sentinel-2 optical image. Finally, the optimal subset regression based on these indices and 55 observations collected from November 1, 2018 to November 11, 2018 was used to estimate the crop residue cover. The 55 observations were measured using the Line-Transect method. The best model shows high accuracy. Because Lishu County has implemented the straw return policy since 2007, the crop residue cover in 2006 can be regarded as 0. The difference between the crop residue cover in 2018 and 2006 was used as one of the variables (CRC) for modelling SOM in 2018 and to evaluate the effects of long-term straw return on SOM variation

during 2006–2018. This study assumed that crop performance or crop types were the same except for CRC. The CRC was used to represent the straw return.

The resolution of land-use types is 30 m in 2005 and 2018. The land-use types in 2005 and 2018 are consistent with six major classes (farmland, woodland, grassland, waters, built-up land, and unused land) and 25 subclasses. The farmland includes upland and paddy land; the woodland includes forestland, shrubbery, open woodland, and other woodland; and the grassland includes high, medium, and low coverage grass land. The land-use data were derived by manual visual interpretation of Landsat TM images. An electronic version of soil type map in Lishu County, offered by the Agricultural Extension Station in Lishu County, was digitized and delineated for this study. The map for soil clay content at 250 m resolution was obtained from SoilGrids250m products (Hengl et al., 2015) in International Soil Reference and Information Center.

The National Earth System Science Data Center, National Science & Technology Infrastructure of China provided annual mean precipitation (AMP) in 2006 and 2018. After multivariate regression analysis of 16 variables (Table 1) and SOM content, the Variance Inflation Factors (VIF) of independent variables were less than five, indicating multicollinearity did not exist among independent variables. These environmental covariates (Table 1) were resampled to 30 m using the bilinear method in ArcGIS 10.2.

Table 1 Environmental covariates used for predicting soil organic matter (SOM).

| Theme | Environmental factors | Original resolution | Source |
| --- | --- | --- | --- |
| Geographical coordinate | Y | 30 m | |
| Terrain | DEM, m | 30 m | http://www.resdc.cn/ |
| | Slope | 30 m | Calculated from DEM |
| | Aspect | 30 m | Calculated from DEM |
| | TR | 30 m | Calculated from DEM |
| | TWI | 30 m | Calculated from DEM |
| | PLC | 30 m | Calculated from DEM |
| | PRC | 30 m | Calculated from DEM |
| | MrVBF | 30 m | Calculated from DEM |
| Vegetation | NDVI (2006) and NDVI (2018) | 30 m | Landsat 5 and Landsat 8 |
| | EVI (2006) and EVI (2018) | 30 m | Landsat 5 and Landsat 8 |
| | Crop residue coverage (2018) | 30 m | Liu et al., (2020) |
| Soil | Land use (2005) and Land use (2018) | 30 m | http://www.resdc.cn/ |
| | Soil type | 1:100, 000 | the Second National Soil Survey |
| | Soil clay content, % | 250 m | https://soilgrids.org/ |

| | | | |
|---|---|---|---|
| Climate | AMP, mm (2006) and AMP, mm (2018) | 1000 m | http://www.geodata.cn/ |

Notes: Y, latitude; DEM, digital elevation model, m; TR, terrain relief index; TWI, topographic wetness index; PLC, plan curvature; PRC: profile curvature; MrVBF, multi-resolution valley bottom flatness; NDVI, normalized difference vegetation index; EVI, enhanced vegetation Index; AMP, annual mean precipitation, mm. The number in parentheses is years of the variable.

## 2.4 Spatial predictive modeling

### 2.4.1 Random Forest (RF)

Many studies have successfully predicted various soil nutrient content by using RF model (Wiesmeier et al., 2011; Guo et al., 2015; Zhang et al., 2017). First proposed by Breiman in 2001, RF (Breiman, 2001) is a tree-based ensemble model. Combined with the idea of feature selection, the approach can increase the diversity of individual decision trees and improve the generalization ability of the final RF model. The RF model combines the predictions of all the individual trees, either by taking the majority vote in the case of classification or averaging the outputs in the case of regression. This approach helps to smooth out the noise in the data and produce more accurate predictions. It includes the number of trees (ntree) and the number of variables available for selection in each split (mtry). We used ten-fold cross-validation to optimize the parameters of RF. The study took SOM content and various environmental factors (Table 1) as the dependent and independent variables to build two RF models: consider all the variables as predictors (RF-all); consider the environmental variables without latitude as predictors (RF- (all-Y)); consider the environmental variables without cumulative crop residue coverage (CRC) as predictors (RF- (all-CRC)).

### 2.4.2 Geographical detector (GE)

GE (Wang and Xu, 2017) is a statistical method used in geographical analysis to identify the factors that contribute to spatial patterns. It is based on the idea that the variation in a dependent variable across a geographical area can be explained by a set of independent variables and their interactions. GE is effective to quantify the spatial heterogeneity of attributes between layers. The method includes factor, interaction, risk, and ecological detector. We used the factor and interaction detector to explore the driving factors for SOM prediction in the study. In the factor detector, the $q$ value (from 0 to 1) and whether it passes the significance test was given. The $q$ value is proportional to the effect of the independent variable on the dependent variable. The interaction detector obtained five different results by comparing the $q$ values ($q1$, $q2$) for each of two factors with the $q$ value ($q3$) of the two factors interaction, as presented in Table 2.

Table 2 Interaction judgment in Geographical detector.

| Judgment | Interaction |
|---|---|
| $q3 < Min (q1, q2)$ | Nonlinear weakening |
| $Min(q1, q2) < q3 < Max(q1)$ | Single factor nonlinear weakening |

| | |
|---|---|
| $q3>$Max $(q1, q2)$ | Double factor enhancement |
| $q3=q1+q2$ | Independent |
| $q3>q1+q2$ | Nonlinear enhancement |

### 2.4.3 Model assessment

Independent verification was used to evaluate the performance of the model. The Lins' Concordance Correlation Coefficient (CC) and the root mean squared error (RMSE) were as evaluation metrics. The data was first randomly divided into a modelling

set and a validation set according to the 7:3 ratio. In the modelling set, ten-fold cross-validation was used to obtain the best parameters of the model through RMSE as an index. The calculation of RMSE and CC is as follows:

$$RMSE = (1/n \times \sum_{i=1}^{n}(p_i - o_i)^2)^{1/2} , \qquad (1)$$

$$CC = 2r\sigma_p\sigma_o/\left(\sigma_p^2 + \sigma_o^2 + (\hat{p}_i - \hat{o}_i)^2\right) , \qquad (2)$$

where $p_i$ and $o_i$ are the predicted value and observed value, respectively. $\hat{p}_i$ and $\hat{o}_i$ are the average value of all predictions

and observations, respectively. $n$ is the number of soil samples. $\sigma_p^2$ is the variances of predicted values and $\sigma_o^2$ is the variances of measured values. r is the correlation coefficient of predictions and observations. Model evaluation, descriptive analysis, and variance analysis were realized in R 4.0.2 (R Core Team, 2020). To satisfy the assumption of data independence, this study randomly selected 1/200 of the total number of pixels using the *sample* function and then conducted pairwise significant difference analyses using the "wilcox.test" method in the *ggviolin* function with the stat_compare_means setting. The RF and

GE models were implemented in the "caret" and "GD" libraries, respectively.

## 3 Results and discussion

### 3.1 Descriptive Statistics of SOM data

As shown in Table 3, from 2006 to 2018, the average SOM content increased from 18.93 g kg$^{-1}$ to 20.84 g kg$^{-1}$, and the coefficient of variations (CV) rose slightly from 0.33 to 0.38. The CVs indicated moderate variation (Cambardella et al., 1994).

The ascending CV could be attributed to more active human activities, such as popularized straw return technology in Lishu County, which is consistent with recent studies (Fan et al., 2020; Hu et al., 2014).

Table 3 Statistical description of SOM in 2006 and 2018 in Lishu County.

| | Year | N | Mean | SD | Min | Max | Skew | Kurtosis | CV |
|---|---|---|---|---|---|---|---|---|---|
| SOM | 2006 | 366 | 18.93 | 6.20 | 6.40 | 39.50 | 0.29 | −0.45 | 0.33 |
| (g kg$^{-1}$) | 2018 | 355 | 20.84 | 7.82 | 2.10 | 64.26 | 0.29 | 1.29 | 0.38 |

Notes: SOM, soil organic matter, g kg$^{-1}$; N, the number of samples; SD, standard deviation; CV, coefficient of variation.

**3.2 Model performance**

Table 4 shows that the validation results considering all the variables as predictors (RF-all) (CC = 0.59, RMSE = 4.54 g kg$^{-1}$, taking 2006 as an example) (Fig. 2) performed better than those considering the environmental variables without latitude as predictors (RF- (all-Y)) did. The result indicated that geographical coordinates were significant for SOM mapping. Compared with a RF-all model to predict nematode worm distribution, Ploton et al. (2020) obtained similar results by using a RF-XY (considering the longitude and latitude as predictors) model because the RF-all model mainly depends on geographic proximity (Van Den Hoogen et al., 2019). In areas dominated by agriculture, agricultural management practices (irrigation, tillage practices, farming systems, and residual management) increased the spatial heterogeneity of SOM, so the spatial distribution of SOM cannot be accurately reflected using individual vegetation indexes or topographic factors. However, few studies integrate these agricultural activities to explain the SOM spatial distribution due to the challenge of collecting them based on remote sensing technology. This study developed RF models with and without CRC. After removing CRC, CC was reduced by 14.9%, and RMSE was increased by 0.23 g kg$^{-1}$ (Table 4). The results demonstrated that the RF model with CRC factor achieved a higher accuracy than the model using common environmental variables alone. Therefore, further research should consider the importance of geographic coordinates and long-term straw return in DSM.

Table 4 The performance of the random forest model.

| Methods | | 2006 | | 2018 | |
|---|---|---|---|---|---|
| | | CC | RMSE (g kg$^{-1}$) | CC | RMSE (g kg$^{-1}$) |
| RF-all | Calibration | 0.55 | 5.04 | 0.44 | 6.99 |
| | Validation | 0.59 | 4.54 | 0.54 | 5.38 |
| RF- (all-Y) | Calibration | 0.50 | 5.22 | 0.38 | 7.18 |
| | Validation | 0.55 | 4.63 | 0.47 | 5.55 |
| RF-(all-CRC) | Calibration | \ | \ | 0.43 | 7.03 |
| | Validation | \ | \ | 0.47 | 5.61 |

Notes: RF-all: consider all the variables as predictors; RF- (all-Y), consider the environmental variables without latitude as predictors; RF-(all-CRC), consider the environmental variables without cumulative crop residue coverage (CRC) as predictors; CC, Lins' Concordance Correlation Coefficient; RMSE, root mean squared error.

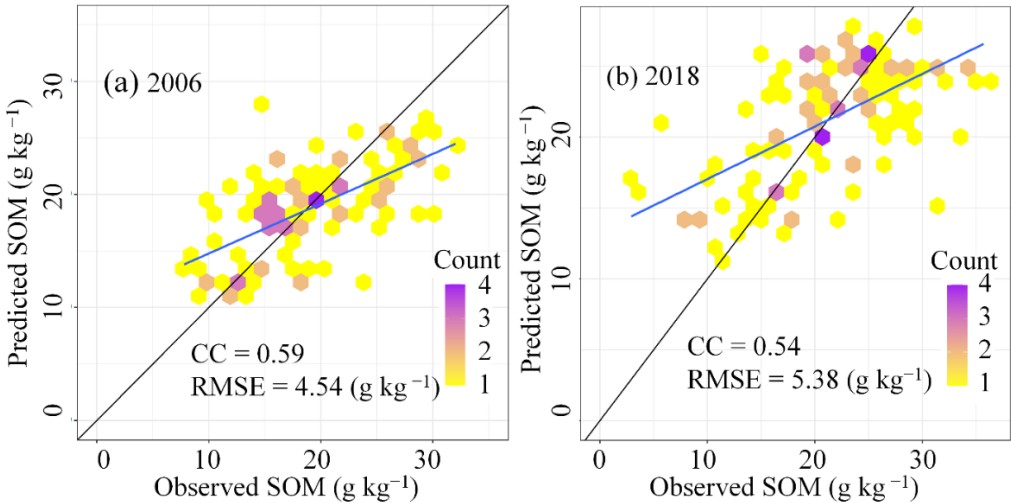

Figure 2: Performances of the random forest with all environmental covariates (RF-all) on validation data in 2006 (a) and 2018 (b). RMSE, root mean squared error; CC, Lins' Concordance Correlation Coefficient; SOM, soil organic matter.

The accuracy of the study's model was not high, which is the study's limitation. By reviewing the SOM or SOC modelling in flat farmland areas, we found that the models of these similar studies all show poor performance (Table 5). This may be because common environmental variables, such as topographic and remote sensing factors, are too homogeneous to effectively extract SOM information (Zeng et al., 2017). Compared with these studies in flat farmland areas, our prediction accuracy is comparable or even better (Table 5).

Table 5 Models' performance in soil organic matter (SOM) prediction in plain farmland areas.

| Study area | Model with common environmental variables | Model performance | References | Characteristics |
|---|---|---|---|---|
| Cultivated land of Xuanzhou city and Langxi County | Random Forest | $R^2 = 0.34$ | Yang et al., 2020 | A typical plain rice production area |
| Chahe Town | Ordinary Kriging; Regression Kriging | CC = 0.15–0.24 | Wu et al., 2021 | A typical plain farmland area |
| Agricultural soils in the north-eastern Iberian Peninsula | General Least Squares | $R^2 = 0.20$–0.27 | Funes et al., 2019 | Agricultural soils |
| Jianghan plain | Stepwise Regression; Partial Least Squares Regression; Extreme learning machine | $R^2 = 0.14$–0.53 | Guo et al., 2021 | Agricultural lands in low-relief areas |

| Miandoab county, West Azerbaijan, northern Iran | Random Forest; Cubist; Conditional Inference Forest; Conditional Inference Trees; Extreme Gradient Boosting; Classification and Regression Trees | CC = 0.34–0.44 | Goydaragh et al., 2021 | The elevation varies from 1292 to 1342 m and the main land use is agriculture |
|---|---|---|---|---|

Notes: CC, Lins' Concordance Correlation Coefficient.

### 3.3 Importance of driving factors on SOM spatial distribution

As Fig. 3a and c prevented, the relative importance of environmental factors obtained from the RF and GE methods was comparable, proving the feasibility of the two methods. For quantifying the relative importance, GE is based on spatial
relationships, while RF is based on feature importance through decision tree splits. The geographical coordinates, soil type, precipitation, clay content, DEM, MrVBF, and CRC play key roles in SOM prediction (the relative importance > 5 %). The relative importance of the other factors was almost less than 5 %. Moreover, these factors passed the significant test in GE model. As Fig. 3b and d presented, the explanatory strength of 16 variables' interaction in pairs was stronger than that of the single factor through nonlinear enhancement or double factor enhancement (Enhancement, bi). This finding proved that the
complex interaction among different influencing factors led to the spatial distribution pattern of SOM. Fig. 3b and d also proved that y, AMP, and DEM have the greatest influence on SOM. Similar to our results, Wang et al. (2017) found that precipitation was the key climatic variable that affects the spatial distribution of SOM in Liaoning, northeastern China. Many studies have revealed the importance of terrain parameters for predicting SOM in Northeast China (Wang et al., 2018; Ma et al., 2017). This may be because DEM-based terrain parameters cause the recombination and redistribution of temperature,
water, light, soil, wind speed, and wind direction, and thus affect the SOM content. At the same time, it shows the CRC was significant using the significance test with GE. CRC is not the main cause of spatial variation of SOM, but the CRC did contribute significantly to the spatial distribution of SOM.

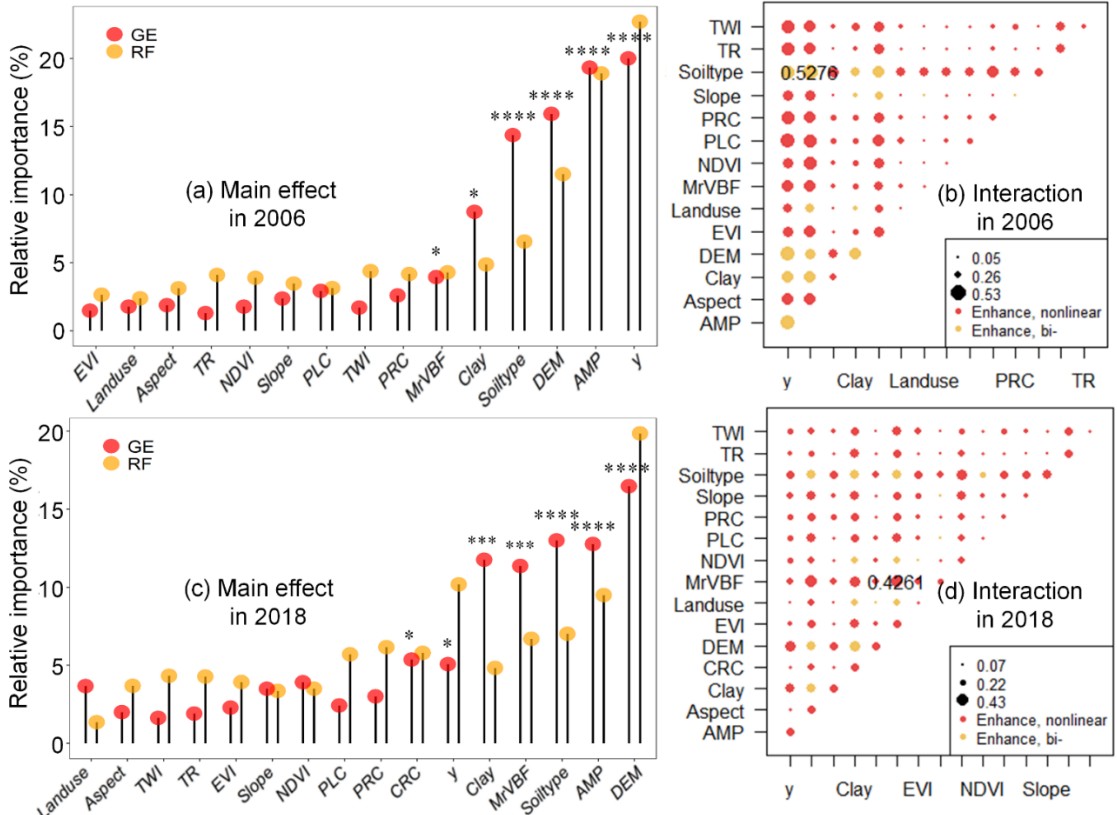

Figure 3: Relative Variables importance (a), (c) and interaction (b), (d) derived from the random forest (RF) and geographic detectors (GE) for soil organic matter (SOM) in 2006 and 2018. Notes: y, latitude; DEM, digital elevation model, m; TR, terrain relief index; TWI, topographic wetness index; PLC, plan curvature; PRC: profile curvature; MrVBF, multi-resolution valley bottom flatness; NDVI, normalized difference vegetation index; EVI, enhanced vegetation Index; CRC, cumulative crop residue coverage; AMP, annual mean precipitation, mm. *, $P < 0.05$; ***, $P < 0.001$; ****, $P<0.0001$

### 3.4 Spatial variation of SOM over time

Fig. 4 presents the spatial distributions of SOM (30 m resolution) in 2006 and 2018. The general spatial pattern was clear: the SOM decreased from southeast to northwest in Lishu County each year. In 2006, SOM concentration was highest in the south of Lishu County (22–28 g kg$^{-1}$) and lowest in the western part (under 12 g kg$^{-1}$). The highest SOM content was concentrated in the middle and east of the study area in 2018, while low SOM content (0–12 g kg$^{-1}$) occupied a tiny area.

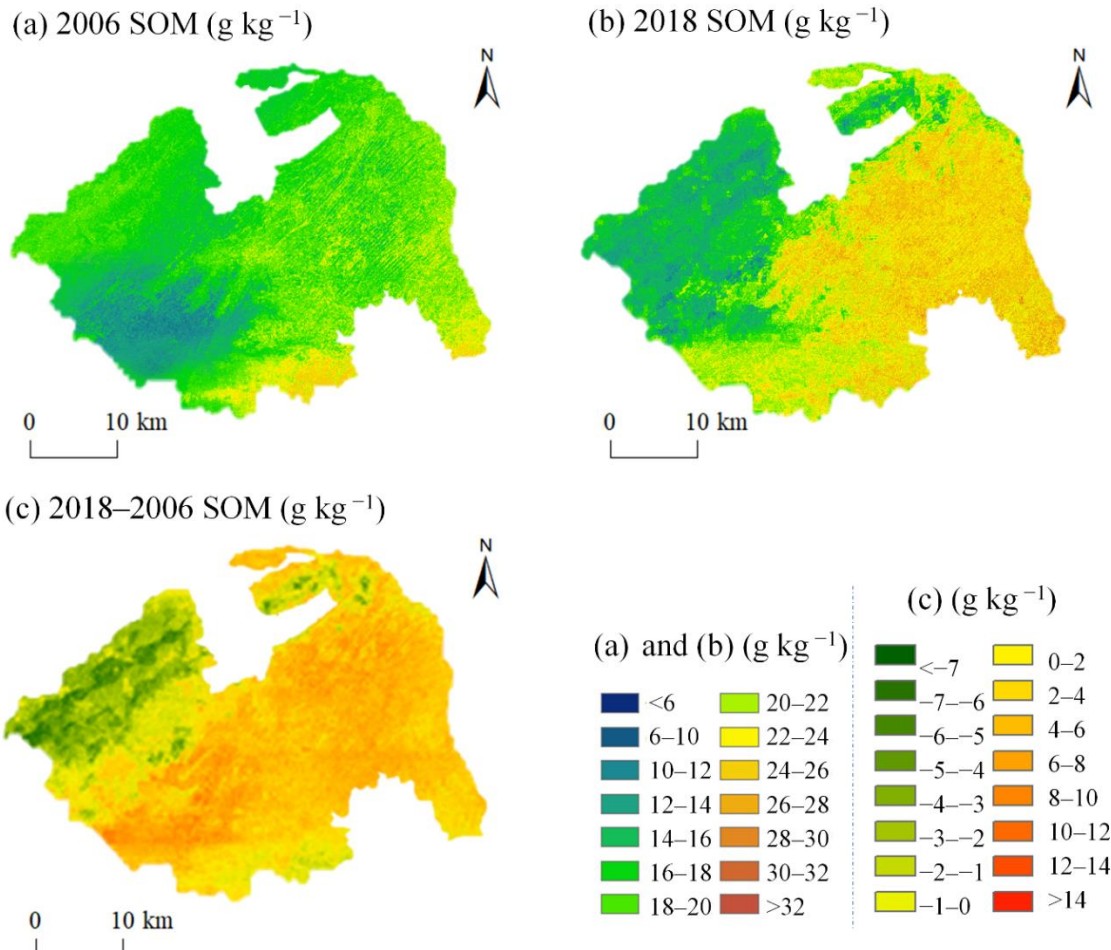

Figure 4: Prediction maps of soil organic matter (SOM) in 2006 (a), 2018 (b), and estimated SOM change between 2006 and 2018 (c) in the study area.

The overall trend of SOM content was on the rise from 2006 to 2018. Consistent with the distribution of high CRC (0.30–1.00) (Fig. 5a), the SOM content in 74.49 % areas of Lishu County displayed a significant increasing trend, especially in the eastern part of the county (Fig. 4c) with a maximum increase of 24.41 g kg$^{-1}$. Many studies have revealed that straw return contributes to carbon sink due to increased microbial biomass and biological activity (Han et al., 2016; Amelung et al., 2020a; Berhane et al., 2020).

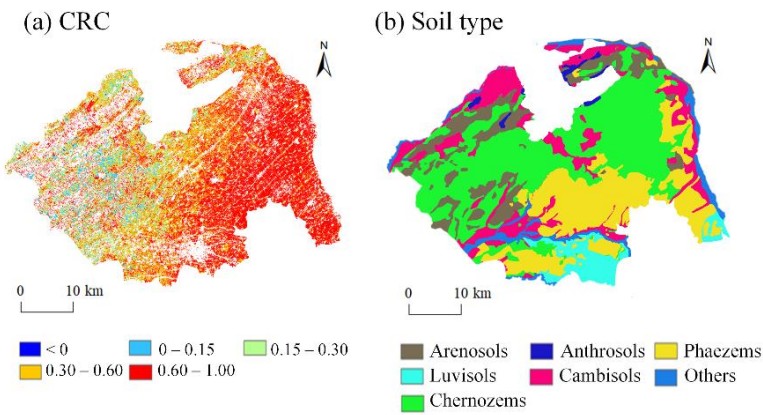

(a) CRC       (b) Soil type

Figure 5: The spatial distribution of cumulative crop residue coverage (CRC) during 2006–2018 (a), and the spatial distribution of soil type (b) in the study area.

The decrease in SOM mainly occurred in the northwest and south of Lishu County. The SOM reduction in southern Lishu County happened to be in the area with the high initial SOM concentration (2006) (Fig. 4c). Consistent with previous research, the reduction was severe in areas with higher initial concentrations of SOC (Zhou et al., 2019). In the northwest corner, the decrease in SOM was mainly distributed on the Arenosols and Anthrosols with a marked decline of 12.70 g kg$^{-1}$ (Figs. 4c, 5b). The phenomenon may be related to the low CRC, light-texture soil, and the change from paddy land to dryland.

## 3.5 Effects of straw return on SOM variation

Fig. 6 presents SOM increased by 0.74, 2.01, 2.68, and 3.08 g kg$^{-1}$ when the CRC was 0–0.15, 0.15–0.30, 0.30–0.60, and 0.60–1.00, respectively. This result proved the SOM increment was proportional to the CRC. A previous study revealed that the effect of straw return on SOM content was closely linked to soil properties, initial SOM content, land-use change, and the straw return amount (Berhane et al., 2020). Therefore, variance analysis was conducted to explore the effects of straw return on SOM variation under different land-use change, soil types, and soil texture.

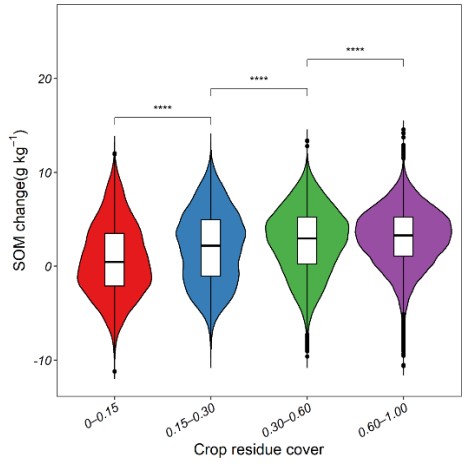

Figure 6: The soil organic matter (SOM) change for different cumulative crop residue coverage (CRC). ****, *P*<0.0001.

### 3.5.1 Effects of the straw return on SOM variation under different soil types

According to Fig. 7, for Anthrosols, Cambisols, and Chernozems, SOM increment was proportionate to the CRC. For Luvisols
and Phaezems, SOM variation showed a decreasing trend with the boosted CRC. This phenomenon is relevant to the initial
SOM contents. As displayed in Fig. 8, the initial SOM contents were lower in Anthrosols, Cambisols, and Chernozems and
higher in Luvisols and Phaezems. Berhane et al. (2020) claimed that regardless of the soil type, the response of SOC change
to carbon input was weak when the initial SOC content was high. Li et al. (2018) concluded that Phaezems with the highest
initial SOM content had the lowest response rate to fertilization. A possible explanation for this might be that soils with low
initial SOM are far away from their saturation levels and thus have a greater potential for carbon sequestration. Except for the
SOM loss in Arenosols and Anthrosols, the SOM changes in other soil types were almost all positive (Fig. 7). This result
verified that the degradation of Arenosols and Anthrosols resulted in the SOM reduction due to the light-texture soil and land-
use change from the paddy land to dryland, respectively.

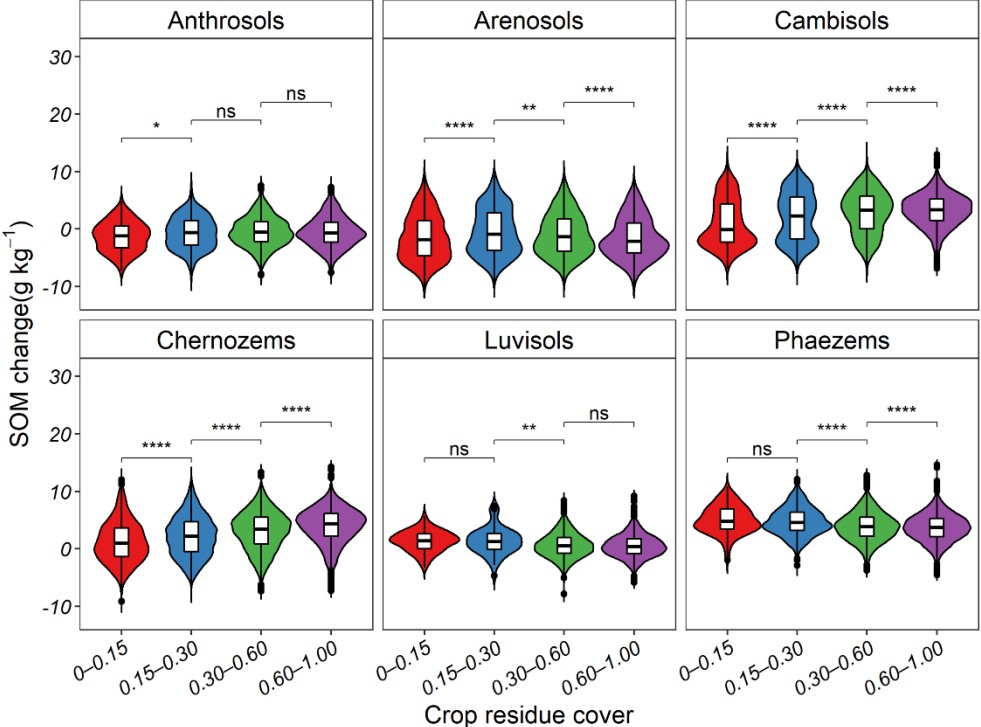

Figure 7: The soil organic matter (SOM) change for different cumulative crop residue coverage (CRC) under soil type. ns,
no significance; *, *P* < 0.05; **, *P* < 0.01; ****, *P*<0.0001.

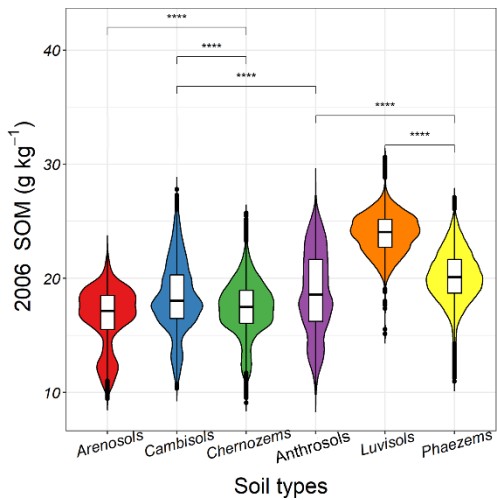

Figure 8: Initial soil organic matter (SOM) content (2006) for soil type. ****, *P*<0.0001.

### 3.5.2 Effects of straw return on SOM variation under different soil texture

With the CRC increasing, SOM first increased and then decreased for soil with low clay content (Fig. 9). Nevertheless, SOM increases with the boosted CRC in soils with high and medium clay contents (Fig. 9), indicating that clay content was directly proportional to the response of SOM increment to CRC. The findings are consistent with previous research, which shows that straw return results in the highest carbon chelation when clay content is high (Xia et al., 2018). One possible explanation for this observation was that soils with higher clay content have greater potential to store organic carbon (Li et al., 2020), so the

SOM accumulation in clayey soils is more responsive to straw return. Moreover, the SOM increment under sandy soil declined when the CRC was 0.60–1.00, which was because sandy soil cannot be protected by mineral particles (Xia et al., 2018) and is more susceptible to the influence of microorganisms.

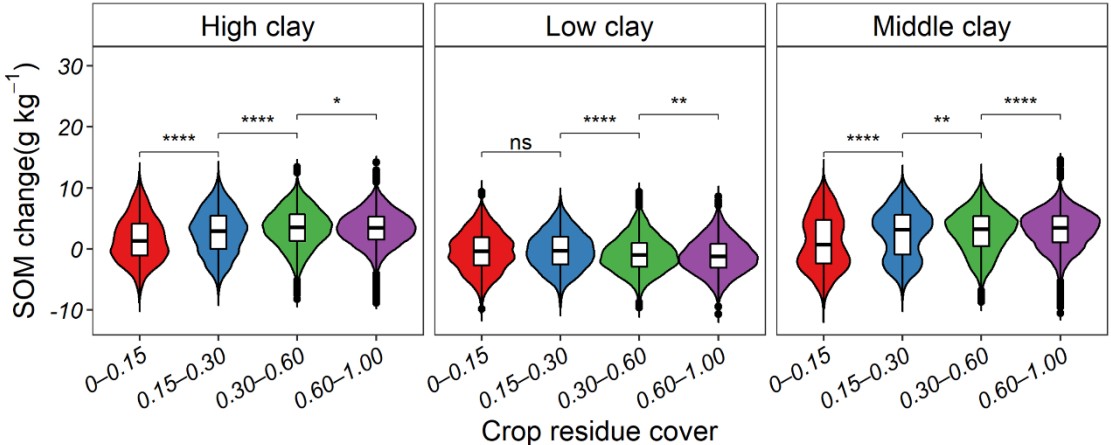

Figure 9: The soil organic matter (SOM) change for different cumulative crop residue coverage (CRC) under soil texture. ns,
no significance; *, *P* < 0.05; **, *P* < 0.01; ****, *P*<0.0001.

### 3.5.3 Effects of straw return on SOM variation under different land-use change

The study considered only two kinds of land-use change: a) the change from dryland to paddy land; b) the change from paddy land to dryland because the crop area accounted for over 70 % of the total sown area in Lishu County and all the samples were taken from farmland. In the two kinds of land-use change, variance analysis was carried out for the SOM variation under four levels of CRC. As shown in Fig. 10, the SOM increment increases with the boosted CRC under the change from dryland to paddy field. From paddy to dry land, the SOM dropped by 0.71 g kg$^{-1}$ when the CRC was 0–0.15. However, with the increase in the CRC, the SOM loss gradually decreased. Even the SOM increased by 1.79 g kg$^{-1}$ when the CRC was 0.60–1.00, indicating that straw return can reverse the carbon loss caused by the transformation of paddy to dryland. Previous studies have found that due to the transformation from anaerobic to aerobic, the conversion of paddy to dry land will cause carbon loss (Wang et al., 2014; Nishimura et al., 2008; Li et al., 2016). Some research pointed out that we can reduce the carbon loss by rewetting (Driessen et al., 2000) or cultivation of paddy fields continuously (Chen et al., 2017). The study pointed out that straw return is a way to prevent a C source caused by the conversion of the paddy field to dryland and can be carried out after that paddy is converted to dryland or when paddy is fallow.

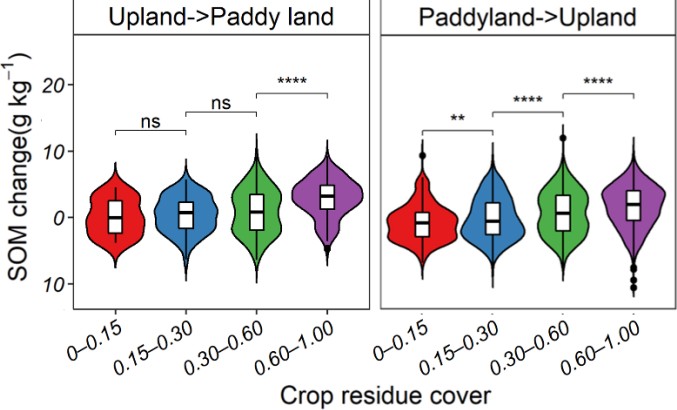

Figure 10: The soil organic matter (SOM) change for different cumulative crop residue coverage (CRC) when paddy land change to dryland. ns, no significance; **, $P < 0.01$; ****, $P<0.0001$.

## 4 Conclusion

The study estimated the surface SOM content and SOM variation between 2006 and 2018 by the RF model, quantified the spatial relationship between measured SOM and predictors, and further explored the effect of the interaction of soil properties, land use, and straw return on SOM change. The results of the study were as follows:

(1) The RF model with all predictors (CC = 0.54, RMSE = 5.38 g kg$^{-1}$ in 2018) performed better than RF-(all-Y) (CC = 0.47, RMSE = 5.55 g kg$^{-1}$) and RF-(all-CRC) (CC = 0.47, RMSE = 5.61 g kg$^{-1}$) in both 2006 and 2018. The result indicated that geographical coordinates and long-term straw return were significant for SOM mapping.

(2) The SOM content decreased from southeast to northwest during both periods. Over time, SOM increased in 74.49% of areas in Lishu County, with the eastern part showing a maximum increase of 24.41 g kg$^{-1}$. The northwest and south corners of the study area aggregated the SOM loss, especially in the northwest, with a significant decline of 12.70 g kg$^{-1}$.

(3) Straw return played the main role in SOM variation. SOM increased by 0.74, 2.01, 2.68, and 3.08 g kg$^{-1}$ with CRC of 0–0.15, 0.15–0.30, 0.30–0.60, and 0.60–1.00. The response rate of SOM to the CRC was inversely proportional to the initial SOM and the sand contents.

The study revealed that straw return is beneficial to carbon sink in farmland and is a better way to prevent a carbon source caused by the conservation of paddy fields to dryland, which can contribute to the development of strategies to ensure the sustainability of agricultural soils.

### Data availability

The data that support the findings of this study are available on request from the corresponding author.

### Author contributions

Conceptualization, Wenjun Ji and Yang Yan; data curation, Guiman Wang and Zhong Liu; methodology, Wenjun Ji and Yang Yan; writing-original draft preparation, Yang Yan; writing-review and editing, Wenjun Ji, Songchao Chen and Dehai Zhu; supervision, Wenjun Ji and Baoguo Li; project administration, Wenjun Ji and Baoguo Li; funding acquisition, Wenjun Ji.

### Competing interests

The authors declare that they have no conflict of interest.

### Financial support

This study was supported by the National Natural Science Foundation of China (Project NO. 42001048), National Key R&D Program of China (2021YFD1500201), Open Fund of State Key Laboratory of Remote Sensing Science (OFSLRSS202121), the State Key Laboratory of Resources and Environmental Information System (2020), and the Chinese Universities Scientific Fund (2020TC205).

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
