# Peer review of "Quantification of the effects of long-term straw return on soil organic matter spatiotemporal variation: A case study in typical black soil region"

_EGUsphere, 2022_

## Author Comment (AC1)

**Responses to Reviewer Comments**

**Dear reviewer:**

Thank you very much for your time involved in reviewing the manuscript, and for providing helpful and specific feedback for how to improve this work. Below we have responded to all of your comments and have indicated how we will change the manuscript as a result of these suggestions.

● **Comment 1**

*As outlined in the title, the main novelty of this manuscript would be the incorporation of straw return amount into the Digital Soil Mapping framework. But descriptions on how the straw return amount (the CRC factor) is quantified and mapped are lacking. The authors did give a reference (Huang et al., 2020; Liu et al. 2020 in Table 1 is not listed in the References) to the data source (I did not manage to get access to the paper), but a more detailed explanation should still be added into the paper. For example, what was the approach used for CRC mapping? What was the sampling design and sample size for CRC field quantification, at what time of year? Without this information, it is difficult to assess the data quality and thus the overall modeling approach.*

● **Response 1**

Thank you very much for the comment.

1) The title of the paper is "Quantification of the effects of long-term straw return on soil organic matter spatiotemporal variation: A case study in typical black soil region". As outlined in the title, the novelty of the paper is the **quantification of the effects of long-term straw return on soil organic matter (SOM) spatiotemporal variation on a regional scale**, but not "the incorporation of straw return amount into the Digital Soil Mapping framework". **As a result, the confounding issues raised by the reviewer don't apply.** We appreciate the comment because our ambiguous writing may have led to the misunderstanding. We guess that the column in the Abstract is ambiguous: "due to lacking accurate spatial distribution of straw return, few studies took straw return as a variable to carry out rigorous research on the impact of straw return on SOM variation on a regional scale", so we corrected the sentences as "due to lacking accurate spatial distribution of straw return, few studies carried out rigorous research on the impact of long-term straw return on SOM spatiotemporal variation on a regional scale" to make the expression clearer and more accurate. Of course, the above is our guess. If there is anything else with a vague expression, please let us know so that we can continue to revise and present the best work to readers.

2) Straw returning is the focus of this paper all the time, so the straw return amount (the CRC factor) should be given a detailed description, as you suggested. We have given a detailed description in Section 2.3 along with the references (Liu, Z., Liu, Z., Wan, W., Huang, J., Wang, J., and Zheng, M.: Estimation of maize residue cover on the basis of SAR and optical remote sensing image, National Remote Sensing Bulletin, 25(06), 1308-1323, 2021. DOI:10.11834/jrs. 20210053):

*"Liu et al. (2020) provided a crop residue coverage at a 10 m resolution in 2018 by combining radar indices and optical remote sensing indices. Firstly, the study divided the study area into a sandy soil area and a clay soil area to reduce the influence of soil properties on radar echo and spectral reflectance. Six radar indices and five optical remote sensing indices were then calculated from a Sentinel-1 SAR image and a Sentinel-2 optical image. Finally, the optimal subset regression based on these indices and 55 observations collected from 1 November 2018 to 10 November 2018 was used to estimate the CRC. The best model shows high accuracy. Lishu County has implemented the straw return policy since 2007. Therefore, the amount of straw return in 2006 can be regarded as 0. The straw returning amount in 2018 minus that in 2016 is numerically equivalent to the straw returning amount in 2018 (CRC), which is used as one of the variables for modelling SOM in 2018 and evaluating the effects of long-term straw return on SOM variation during 2006–2018.*"

3) At the same time, we would like to thank the reviewer again for this comment, which gave us some inspiration. We decided to conduct additional experiments in this study to evaluate the incorporation of straw return amount into the Digital Soil Mapping framework (please see Response 3).

- **Comment 2 (1)**

*2. I think this is problematic because the effect of organic inputs on SOM dynamics is a mid-to-long term process, so linking spatial variability of SOM to CRC at one timepoint seems a bit farfetched to me. I would suggest the authors to look into the cumulative effect of straw return on SOM over a longer time period.*

- **Response 2 (1)**

Thank you very much for the comment.

In this study, the CRC data is mainly used in two aspects: 1) to assess the impact of long-term straw return on soil organic matter variation; 2) to evaluate the importance of CRC in the spatial modeling of SOM in 2018. In both aspects, we used the value of **the straw returning amount in 2018 minus that in 2016** (Figure 5 (a)), which is the cumulative straw return amount. We appreciate the comment because although we provided the information in lines 88-89, it was not clearly stated: "Lishu County has implemented the straw return policy since 2007. Therefore, the amount of straw return in 2006 can be regarded as 0". So, we added more information in this section: "Lishu County has implemented the straw return policy since 2007. Therefore, the amount of straw return in 2006 can be regarded as 0. The straw returning amount in 2018 minus that in 2016 is numerically equivalent to the straw returning amount in 2018, which is used as one of the variables for modelling SOM in 2018 and evaluating the effects of long-term straw return on SOM variation." In addition, we've clarified our wording in the Notes of Table 1, Figure 3, Section 3.2 (291), and Section 3.3 (337).

- **Comment 2 (2)**

*As far as I understood, the authors only used the CRC data in 2018 to evaluate the effect of straw return on SOM content in the same year.*

- **Response 2 (2)**

Thank you very much for the useful comment. Your suggestion is valid. At present, there is no efficient method to investigate the cumulative effect of straw return on SOM spatiotemporal variation on a regional scale. As you pointed out, this study should utilize the cumulative straw returning amount several years ago (such as, 2015-2006). But we can only collect the spatial distribution data of straw returning in 2018. Based on the data available, this is the most efficient approach we can take.

- **Comment 3**

*3. For the RF model, the authors included the CRC factor for the year 2018, but the relative importance of CRC appears to be low. This again questions the validity of the approach and the CRC data. At the very least, the authors should compare the predictive performances of models with and without CRC, so as to demonstrate whether incorporating CRC improves the model performance.*

- **Response 3**

Thank you for the comment.

In order to avoid misunderstanding, we must emphasize that, as mentioned above, the focus of this paper is to evaluate the impact of long-term straw returning on soil organic matter change. The spatial modeling in 2018 was calibrated to obtain the spatial distribution of organic matter only, and it is not the focus. Although the relative importance of CRC in the RF nodel is not the focus of this article, as mentioned in comment 1, we thank you for the comment, because it provides us with new ideas, and this article can explore the significance of straw returning in the DSM. So, we will add the content about this idea according to your suggestion. As you suggested, we added a set of experiments. We established RF models with CRC and without CRC and found that, after removing CRC, CC was reduced by 14.9%, and RMSE was increased by 0.23 g kg$^{-1}$ (Table 4). We have added the result and the related discussion in section 3.2: "*In areas dominated by agriculture, agricultural management practices (irrigation, tillage practices, farming systems, and residual management) increased the spatial heterogeneity of SOM, so the spatial distribution of SOM cannot be accurately reflected using individual vegetation indexes or topographic factors. However, few studies integrate these agricultural activities to explain the SOM spatial distribution due to the challenge of collecting them based on remote sensing technology. This study developed RF models with and without CRC. After removing CRC, CC was reduced by 14.9%, and RMSE was increased by 0.23 g kg$^{-1}$ (Table 4). The results demonstrated that the RF model with the CRC factor achieved a higher accuracy than the model using common environment variables alone. Therefore, further research should consider the*

*importance of geographic coordinates and long-term straw return in DSM."*

Table 1 The performance of the random forest model.

| Methods | | 2006 | | 2018 | |
|---|---|---|---|---|---|
| | | CC | RMSE (g kg$^{-1}$) | CC | RMSE (g kg$^{-1}$) |
| RF-all | Calibration | 0.55 | 5.04 | 0.44 | 6.99 |
| | Validation | 0.59 | 4.54 | 0.54 | 5.38 |
| RF- (all-Y) | Calibration | 0.50 | 5.22 | 0.38 | 7.18 |
| | Validation | 0.55 | 4.63 | 0.47 | 5.55 |
| RF-(all-CRC) | Calibration | \ | \ | 0.43 | 7.03 |
| | Validation | \ | \ | 0.47 | 5.61 |

To ensure the rigor of the research, RF and GE methods were used to analyze the importance factors, as shown in the figure, CRC ranked the seventh in this study. At the same time, it shows the CRC was significant using the significance test with GE. Straw returning is not the main cause of spatial variation of SOM (of course, topography, climate and soil type are the main contributions), but the CRC did contribute significantly in the spatial distribution of SOM.

[Figure]

- **Comment 4**

*4. The predictive performances (CC and RMSE) for the 2006 and 2018 RF models were not exceptional (Figure 2) – the accuracy was actually worse in 2018 after the addition of CRC. This means that the predicted SOM values are associated with large prediction errors and uncertainties, thus weakening the obtained results from direct comparisons for the purpose of SOM monitoring.*

- **Response 4**

Thank you for the comment.

As shown in the additional experiments (Table 1), when CRC was removed from the RF model in 2018, CC was reduced by 14.9%, and RMSE was increased by 0.23 g kg[-1]. So the accuracy was higher in 2018 after the addition of CRC. For the comment "the accuracy was actually worse in 2018 after the addition of CRC", we're a little unclear what the reference is.

The modeling accuracy is the study's limitation. In general, the modeling accuracy of SOM or SOC in other similar studies (county-scale, and the study area characterized with plat farmland area) is low (Table 2), which may be because topographic and remote sensing factors are too homogeneous to effectively extract SOM information. The improvement of modeling accuracy for SOM or SOC in county-scale with similar characteristics is indeed a challenge. However, compared with similar studies on flat agricultural areas, the model of our study performed better. This content was added in Section 3.2: "*The accuracy of the study's model was not high, which is the study's limitation. By reviewing the SOM or SOC modelling in flat farmland areas, we found that the models of these similar studies all show poor performance (Table 5). This may be because common environmental variables, such as topographic and remote sensing factors are too homogeneous to effectively extract SOM information (Zeng et al., 2017). Compared with these studies in flat farmland areas, our prediction accuracy is comparable or even better (Table 5).*"

Table 2 Models' performance in SOM prediction in plain farmland areas.

| Study area | Model with common environmental variables | Model performance | References | Characteristics |
|---|---|---|---|---|
| Cultivated land of Xuanzhou city and Langxi County | Random Forest | $R^2$=0.34 | Yang et al., 2020 | A typical plain rice production area |
| Chahe Town | Ordinary Kriging; Regression Kriging | CC=0.15-0.24 | Wu et al., 2021 | A typical plain farmland area |
| Agricultural soils in the north-eastern Iberian Peninsula | General Least Squares | $R^2$=0.20-0.27 | Funes et al., 2019 | Agricultural soils |
| Jianghan plain | Stepwise Regression; Partial Least Squares Regression; Extreme learning machine | $R^2$=0.14-0.53 | Guo et al., 2021 | Agricultural lands in low-relief areas |
| Miandoab county, West Azerbaijan, northern Iran | Random Forest; Cubist; Conditional Inference Forest; Conditional Inference Trees; Extreme Gradient Boosting; Classification and Regression Trees | CC=0.34-0.44 | Goydaragh et al., 2021 | The elevation varies from 1292 to 1342 m and the main land use is agriculture |

Funes, I., Savé, R., Rovira, P., Molowny-Horas, R., Alcañiz, J. M., Ascaso, E., Herms, I., Herrero, C., Boixadera, J., and Vayreda, J.: Agricultural soil organic carbon stocks in the north-eastern Iberian Peninsula: Drivers and spatial variability, Science of the Total Environment, 668, 283-294, 2019.

Goydaragh, M. G., Taghizadeh-Mehrjardi, R., Jafarzadeh, A. A., Triantafilis, J., and Lado, M.: Using environmental variables and Fourier Transform Infrared Spectroscopy to predict soil organic carbon, Catena, 202, 105280, 2021.

Guo, L., Fu, P., Shi, T., Chen, Y., Zeng, C., Zhang, H., and Wang, S.: Exploring influence factors in mapping soil organic carbon on low-relief agricultural lands using time series of remote sensing data, Soil and Tillage Research, 210, 104982, 2021.

Wu, Z., Liu, Y., Han, Y., Zhou, J., Liu, J., and Wu, J.: Mapping farmland soil organic carbon density in plains with combined cropping system extracted from NDVI time-series data, Science of The Total Environment, 754, 142120, 2021.

Yang, L., He, X., Shen, F., Zhou, C., Zhu, A. X., Gao, B., Chen, Z., and Li, M.: Improving prediction of soil organic carbon content in croplands using phenological parameters extracted from NDVI time series data, Soil and Tillage Research, 196, 104465, 2020.

**Specific comments:**

● **Comment 5**

*1. In the Abstract, one should briefly mention the size and characteristics of the study area.*

● **Response 5**

Thank you for your useful recommendation. We have added these information in the Abstract: "*This study was carried out across an approximately 3000 km² area in Lishu County, Northeast China, a typical agricultural plain.*"

● **Comment 6**

*2. Line 35-40, what is the difference between conventional mapping and DSM? Doesn't DSM also comprise the procedures you outlined in the first sentence of the paragraph?*

● **Response 6**

Thank you for your comment. Comparing digital soil mapping to traditional soil mapping, the most notable difference is that digital soil mapping makes use of quantitative inference models to provide predictions of soil properties in a geographical database (raster) (https://www.nrcs.usda.gov/resources/data-and-reports/digital-soil-mapping-dsm). We originally intended to express the figure below, but the expression has a problem. We corrected the sentence "*Conventional mapping involves data collection, field investigation, interpretation, field inspection, calibration, and mapping.*" as: "*Conventional mapping involves laboriously constructing maps by planetable and alidade (Ahrens, 2008).*".

Ahrens, R. J.: Digital soil mapping with limited data, Springer Science & Business Media, 2008.

[Figure]

- **Comment 7**

*3. Line 70-75, there is no mention of the sampling designs in 2006 and 2018. Also, what are the sample sizes?*

- **Response 7**

Thank you for your useful recommendation. We have added these information in Section 2.3: "*By taking into account the sample sites in the second national soil survey, local landform, and soil types, a total of 300 sampling sites in 2006 and 319 sampling sites in 2018 were selected. The soil samples were collected on the surface (0–20 cm) from early October to mid November in each year (from the harvest to the freezing)*"

- **Comment 8**

*4. Line 85, I suggest the authors to specify how NDVI and EVI were calculated? Annual mean or based on images from a specific month?*

- **Response 8**

Thank you for your useful recommendation. We have added these information in Section 2.3: "*The average reflectance of three image bands (B1 (Blue), B3 (Red) and B4 (Near-infrared)) of Landsat 5 SR and Landsat 8 SR products spanning May to September were processed to calculate the NDVI by $(B4 - B3)/(B4 + B3)$ and EVI by $2.5 \times (B4 - B3)/(B4 + 6 \times B3 - 7.5 \times B1 \times 1)$. The band calculation, and image clipping were conducted in Google Earth Engine (GEE), and the images with less than 6% of cloud coverages were selected.*"

- **Comment 9**

*5. The authors should specify the statistical method used for significance tests for all the boxplots. Otherwise, it is difficult to evaluate the appropriateness of the comparisons on changes in SOM with varying straw return amount.*

- **Response 9**

Thank you for your commet. The statistical method used for significance tests for all the boxplots is wilcoxon test , which is realized in "stat_compare_means" in R 4.0.2. The information has been added in Section 2.4.3: "*The statistical method used for significance tests for all the boxplots is Wilcoxon test, which is realized in "stat_compare_means". The RF and GE models were implemented in the "caret" and "GD" libraries, respectively.*"

- **Comment 10**

*6. The entire Results and discussion section was more focused on the interpretation of the results. An in-depth discussion on the strengthens and weaknesses of the methodology is missing.*

- **Response 10**

Thank you for your useful recommendation. As your suggestion, we have added the information in Section 3.2: "*The accuracy of the study's model was not high, which is the study's limitation. By reviewing the SOM or SOC modelling in flat farmland areas, we found that the models of these similar studies all show a poor performance (Table 5). This may be because common environmental variables, such as topographic and remote sensing factors are too homogeneous to effectively extract SOM information (Zeng et al., 2017). Compared with these studies in flat farmland areas, our prediction accuracy is comparable or even better (Table 5).*"

- **Comment 11**

*7. Overall, the writing of the manuscript should be improved.*

- **Response 11**

Thank you for your useful recommendation. We have asked colleague who speacks English to carefully check, and we will improve the English writing in the revised manuscript.

---

## Author Comment (AC2)

**Dear Reviewer 1:**

Thank you very much for your time involved in reviewing the manuscript and your very encouraging comments on the merits.

- **Comment 1**

*The introduction can and should be expanded by some discussion on the mechanism of the influence of straw return on soil organic matter, it is necessary to include more references. Also, there are more than 6 soil types in the study area. but this paper only compared SOM results in 6 soil types in the section "Results and discussion".*

- **Response 1**

Thank you for the useful comment. In "Introduction" part, we did not involve discussion on the mechanism of the influence of straw return on soil organic matte. Based on your opinions, we read more papers and added some contents in the section:  "Straw return is beneficial for retaining soil moisture and preventing soil wind erosion, especially in the arid and semi-arid region. In addition, the decomposition process of straw promotes the activity of microorganisms and is conducive to SOM accumulation (Chang et al., 2014; Lu et al., 2009; Wang et al., 2015). Conversely, previous scholars have reported that the influence of straw return on SOM accumulation is non-significant (Pittelkow et al., 2015; Poeplau et al., 2015; Powlson et al., 2011). This may be because adding organic matter to the soil has no effect on its chemical, chemical and biological properties (Sosulski et al., 2011) or this practice may contribute to the SOM mineralization process and thus reduce SOM (Šimanský et al., 2019)." Your suggestion has greatly improved the level of the article and also brought some new inspirations for the future idea. Thank you again for your valuable suggestions

This paper only compared SOM results in 6 soil types was because Arenosols, Cambisol, Chernozenms, Luvisols, Phaezems, and Anthrosols are the main soil type in the study area. The other soil types' area was small. Therefore, other soil types were not considered in the study.

- **Comment 2**

*Line 47-48, references should be added*

- **Response 2**

Thank you for the detailed comment. We have added references.

- **Comment 3**

*Line 53, remove "of three folds" and change to "3-fold"*

- **Response 3**

Thank you for your detailed comment. We have changed "of three folds" to "3-fold".

- **Comment 4**

*Line 54, add the full name of "RF"*

- **Response 4**

Thank you for your recommendation. We have added the full name of "RF".

- **Comment 5**

*Line 90, land-use types. This item needs more description.*

- **Response 5**

Thank you for your detailed comment. We have added more description about the land-sue types: "The resolution of land-use types is 30 m in 2005 and 2018. The land-use types in 2005 and 2018 are consistent with six major classes (farmland, woodland, grassland, waters, built-up land, and unused land) and 25 subclasses. The farmland includes upland and paddy land, the woodland includes forestland, shrubbery, open woodland, and other woodland, and the grassland includes high, medium and, low coverage grass land. The land-use data were derived by manual visual interpretation of Landsat TM images."

- **Comment 6**

*Line 145, needs an explanation of "RF-XY"*

- **Response 6**

Thank you for your detailed comment. We have added it.

- **Comment 7**

*Figure 2: Missing text of X-axis*

- **Response 7**

Thank you. We have added the text of X-axis in Figure (b)

- **Comment 8**

*In Figure 6,7,8,9,10, please explained "***"*

- **Response 8**

Thank you for your detailed comments. We have corrected it.

- **Comment 9**

*Table 1 If the year is not marked, are the variables used in two years?*

- **Response 9**

Thanks for your advice. We have expalined it in Notes of Table 1:

- **Comment 10**

*A little more discussion in section 3.3 could also help readers.*

- **Response 10**

Thank you for the comments. We have added discussion in section 3.3: "Similar to our results, Wang et al. (2017) found that precipitation were the key climatic variables that affect the spatial distribution of SOM in Liaoning, northeastern China. Many studies revealed the importance of terrain parameters for predicting SOM in Northeast China (Wang et al., 2018; Ma et al., 2017). This may be because DEM-based terrain parameters cause the recombination and redistribution of temperature, water, light, soil, wind speed and wind direction, and thus affect the SOM content."

- **Comment 11**

*Unified "straw return" or "straw returning"*

- **Response 11**

Thank you for your detailed comments. We have unified as "straw return" for the whole text.

---

## Author Comment (AC4)

Dear Reviewer 3:

Thank you very much for your time involved in reviewing the manuscript and your very useful comments. This feedback greatly improved the quality of our paper and made this article more rigorous.

Comment: The manuscript addresses an important topic: the effect of straw return on soil organic matter. However, the introduction is rather short and does not fully explain the spatio-temporal modelling of soil properties.

Response: Thank you for your comment. We appreciate your comments and agree that the introduction could be more detailed in explaining the spatio-temporal modelling of soil properties. We have revised the introduction to provide more information on the spatio-temporal modelling of soil properties, including a brief explanation of the methods used to model soil properties:"Based on the soil-forming theory, digital soil mapping (DSM) uses statistical and geospatial techniques to model the relationship between soil properties and environmental covariates at a high spatial resolution. By analyzing the relationships between soil properties and environmental factors, DSM models can be developed to predict the soil properties of areas where no soil data exist. Therefore, it offers a promising solution for predicting soil properties with high precision and tremendous speed (Hengl et al., 2015; Dou et al., 2019; Liang et al., 2019; Schulze and Schütte, 2020). Moreover, DSM can also incorporate the temporal component in soil property mapping by taking time as an index and comparing soil maps at two moments to identify changes in soil properties over time. This is particularly useful in understanding the impact of land use and management practices on soil properties and identifying areas where remediation may be necessary." We have also included additional information on the mechanism of straw returning affecting SOM:"Straw return is beneficial for retaining soil moisture and preventing soil wind erosion, especially in arid and semi-arid regions. In addition, the decomposition process of straw promotes the activity of microorganisms and is conducive to SOM accumulation (Chang et al., 2014; Lu et al., 2009; Wang

et al., 2015). Conversely, previous scholars have reported that the influence of straw return on SOM accumulation is non-significant (Pittelkow et al., 2015; Poeplau et al., 2015; Powlson et al., 2011). This may be because adding organic matter to the soil has no effect on its chemical, chemical, and biological properties, (Sosulski et al., 2011) or this practice may contribute to the SOM mineralization process and thus reduce SOM (Šimanský et al., 2019)."

Comment: Furthermore, the materials and methods lack precision and the protocols deviate from current practices. Why was the soil sieved at 0.25 mm, while the fine earth is generally defined as < 2 mm (line 78). There is also a confusion between SOM and SOC. The wet oxidation protocol should be explained more carefully, because these analyses determine the SOC and NOT the SOM content.

Response: Thank you for your comment. I understand your concerns regarding the precision of the materials and methods, as well as the confusion between SOM and SOC. Regarding the soil sieving, I agree that there may be some confusion in the manuscript. After air drying and grinding, the soil samples were thoroughly mixed and passed through a 2 mm mesh. The samples were ground and sieved to separate a particle size fraction (0.25 mm) to determine SOC concentration. I reviewed the materials and methods section to ensure that this was clearly explained.

As for the confusion between SOM and SOC, I understand that the wet oxidation method is used to determine SOC content. Specifically, it is the external heating potassium dichromate volumetric method, which can then be multiplied by a conversion factor of 1.724 to obtain the SOM amount. Because some studies have written that SOM was measured using external heating potassium dichromate volumetric method (Lu et al., 2022; Ma et al., 2022), we continued to write it this way. We have revised this in Section 2.2:

"Sun exposure, acid, alkali, and dust pollution were strictly prohibited. After air drying and grinding, the soil samples were thoroughly mixed and passed through a 2 mm mesh. The samples were ground and sieved to separate a particle size fraction (0.25 mm) to determine SOC concentration

with the external heating potassium dichromate volumetric method, which can then be multiplied by a conversion factor of 1.724 to obtain the SOM amount (Liu et al., 1996)."

Lu, M. Y., Liu, Y., & Liu, G. J. (2022). Precise prediction of soil organic matter in soils planted with a variety of crops through hybrid methods. Computers and Electronics in Agriculture, 200, 107246.
Ma, R., Hu, F., Xu, C., Liu, J., & Zhao, S. (2022). Response of soil aggregate stability and splash erosion to different breakdown mechanisms along natural vegetation restoration. Catena, 208, 105775.

Comment: the effect of straw return (section 3.5) is difficult to evaluate. First of all, the term is not clearly defined. According to the materials and methods section it is the residue cover and not the percentage of the residue produced by the crop.

Response: Thank you very much for your feedback. I agree that the term "straw return" in our study is the crop residue cover. We have clearly defined it in the section 2.3. "Crop residue cover refers to the ratio of the vertical projected area of crop residue in a field per unit area to the total surface area of this unit area, with value ranging from 0 to 1." In addition, we have modified this term in the whole text, including the title, figures, and tables to ensure readers understand the intended meaning. At the same time, we have added more details about the CRC in Section 2.3, including how to measure straw coverage: "Liu et al. (2020) provided a crop residue coverage map at a 10 m resolution in 2018 by combining radar indices and optical remote sensing indices. Crop residue cover refers to the ratio of the vertical projected area of crop residue in a field per unit area to the total surface area of this unit area, with value ranging from 0 to 1. Firstly, the study divided the study area into a sandy soil area and a clay soil area to reduce the influence of soil properties on radar echo and spectral reflectance. Six radar indices and five optical remote sensing indices were then calculated from a Sentinel-1 SAR image and a Sentinel-2 optical image. Finally, the optimal subset regression based on these indices and 55 observations collected from November 1, 2018 to November 11, 2018 was used to estimate the crop residue cover. The 55 observations were measured using the Line-Transect method (Wollenhaupt and Pingry, 1991). The best model shows high accuracy."

Comment: Second, the statistical analysis of the effect (see Fig. 6) is poorly explained. If Fig. 6

displays the SOM content of the pixels in each class, these observations are not independent and therefore cannot be pair wise compared using a statistical test (ANOVA or t test). This remark also holds for figures 7-10.

Response: Thank you very much for your comment. To improve clarity and ensure consistency, we have revised figures 6, 7, 9, 10 to match the style of Fig. 8. Because some data did not meet the assumptions of normal distribution, we used the Wilcoxon signed-rank test (a non-parametric statistical hypothesis test) to compare the means of two groups. To satisfy the assumption of data independence, we randomly selected 1/200 of the total number of pixels using the *sample* function in R 4.0.2. Using the impact of straw incorporation on soil organic matter as an example, we categorized the crop residue cover into four levels (1: 0–0.15; 2: 0.15–0.30; 3: 0.30–0.60; 4: 0.60–1.00). We ordered these groups by the mean SOM values of each level and then conducted pairwise significant difference analyses using the "wilcox.test" method in the ggviolin function with the stat_compare_means setting for (1,2), (2,3), and (3,4) levels. The code is as follows:

my_comparisons <- list( c("1", "2"), c(2","3"),c("3","4"))

f%>%ggviolin(x="CRC",y="X2018.2006",fill="CRC",add="boxplot",add.params=list(fill="white" )) +stat_compare_means(comparisons = my_comparisons,method="wilcox.test", label = "p.signif") ->p

We have added this information in section 2.4.3.

Lines 35-36 Would not it be better to express the functions of Jenny and SCORPAN with the dependent variable 'soil property' rather than 'soil'. After all, 'soil' is a broad concept that cannot be quantified and you mention 'soil properties ' in line 38.

Thank you for your comment. We have revised it as you suggested.

 Lines 40-43 You have explained (not in great detail) the role of DSM for quantifying the spatial variation in soil properties. Here you also include the temporal component. This has to be explained

in more detail.

Thank you for your comment. We appreciate your comments and agree that the introduction could be more detailed in explaining the spatio-temporal modelling of soil properties.

We have revised the introduction to provide more information on the spatio-temporal modelling of soil properties, including a brief explanation of the methods used to model soil properties: "Based on the soil-forming theory, digital soil mapping (DSM) uses statistical and geospatial techniques to model the relationship between soil properties and environmental covariates at a high spatial resolution. By analyzing the relationships between soil properties and environmental factors, DSM models can be developed to predict the soil properties of areas where no soil data exist. Therefore, it offers a promising solution for predicting soil properties with high precision and tremendous speed (Hengl et al., 2015; Dou et al., 2019; Liang et al., 2019; Schulze and Schütte, 2020). Moreover, DSM can also incorporate the temporal component in soil property mapping by taking time as an index and comparing soil maps at two moments to identify changes in soil properties over time. This is particularly useful in understanding the impact of land use and management practices on soil properties and identifying areas where remediation may be necessary."

Lines 74 and 75 The sampling design and use of legacy data is not discussed, so it is difficult to interpret their effects on 'prediction error'.

Thank you for your comment. We have provided more details on the sampling design: "**By taking into account the sample sites in the second national soil survey, local landform, and soil types, a total of 300 sampling sites in 2006 were selected. Except for considering these factors, grid sampling was combined to select 319 sampling sites in 2018.** The soil samples were collected on the surface (0–20 cm) from early October to mid-November in each year (from the harvest to the freezing). The corresponding longitude and latitude were also documented. The prediction error caused by the differences in sampling designs for the years 2006 and 2018 was not considered to make full use of legacy soil data" to improve the readers' understanding of the impact on prediction

error. We use legacy data from both 2006 and 2018. The type and source of the legacy data are the same for both years.

Line 82 Please provide the reference for the 'Resource and environment data cloud platform'

Thank you for your comment. We have provided a website for the 'Resource and environment data cloud platform' : "A 30 m resolution digital elevation model (DEM) was derived from the Resource and Environment Data Could Platform (**http://www.resdc.cn/)**."

Line 87 Although spectral indices such as NDVBI and EVI are well-known, this is much less the case for the NDTI and STI. Please specify these indices.

Thank you for your comment. We have specified all the vegetation indices and described their calculation formulas in this paper.

Line 89 If I understand correctly, you use the CRC of 2018 for all fields between 2007 and 2018? This is a strong assumption as it does not take differences in crop performance or crop type into account. Please describe more clearly that the CRC is not used as a co-variate, but you compare two datasets (with and without residue).

Thank you for your comment. We have described more clearly the use of the CRC in Section 2.3: "Because Lishu County has implemented the straw return policy since 2007, the crop residue cover in 2006 can be regarded as 0. The difference between the crop residue cover in 2018 and 2006 (CRC) was used as one of the variables for modelling SOM in 2018 and to evaluate the effects of long-term straw return on SOM variation during 2006–2018. This study assumed that crop performance or crop types were the same except for CRC. The CRC was used to represent the straw return."

Section 2.4.2 The technique of geographical detector is not as widely known as e.g. random forest.

The principles will have to explained in a couple of sentences.

Thank you for your comment. We have added a couple of sentences to explain the principles of GE: "GE (Wang and Xu, 2017) is a statistical method used in geographical analysis to identify the factors that contribute to spatial patterns. It is based on the idea that the variation in a dependent variable across a geographical area can be explained by a set of independent variables and their interactions."

Table 3 Please include a column with the number of samples.

Thank you for your comment. We have included a column with the number of samples.

Section 3.2 Please explain the abbreviations e.g. '(all-Y)'. As it stands the reader has to look them up in the figure caption. It is not clear either whether these statistics apply to the calibration or the validation data set.

Thank you for your comment. We have included a more comprehensive explanation of the abbreviations in the text of the manuscript: "Table 4 shows that **the validation results considering all the variables as predictors** (RF-all) (CC = 0.59, RMSE = 4.54 g kg$^{-1}$, taking 2006 as an example) (Fig. 2) performed better than **that considering the environment variables without latitude as predictors (RF- (all-Y))** did." The statistics presented in Section 3.2 refer to the performance of the model on the validation dataset. We have also added this information to make it clearer.

Line 180 and further on. Please define what you mean by 'straw return content'. As far as I can see it is the straw cover and not necessarily the percentage of residues produced.

Thank you very much for your feedback. I agree that the term "straw return" in our study is the crop residue cover. We have clearly defined it in Section 2.3. "Crop residue cover refers to the ratio of the vertical projected area of crop residue in a field per unit area to the total surface area of this unit

area, with value ranging from 0 to 1." In addition, we have modified this term in the whole text, including the title, figures, and tables, to ensure readers understand the intended meaning.

Figure 6 Please explain how the significance was calculated and what it means. Why did you not try to fit a regression and analyse the significance of the regression?

Using the impact of straw incorporation on soil organic matter as an example, we categorized the crop residue cover into four levels (1: 0–0.15; 2: 0.15–0.30; 3: 0.30–0.60; 4: 0.60–1.00). We ordered these groups by mean SOM values of each level and then conducted pairwise significant difference analyses using the "wilcox.test" method in the ggviolin function with the stat_compare_means setting for (1,2), (2,3), and (3,4) levels. The code is as follows:

my_comparisons <- list( c("1", "2"), c(2","3"),c("3","4"))

f%>%ggviolin(x="CRC",y="X2018.2006",fill="CRC",add="boxplot",add.params=list(fill="white" )) +stat_compare_means(comparisons = my_comparisons,method="wilcox.test", label = "p.signif") ->p

We have added this information in Section 2.4.3.

We did not conduct regression analysis because each level of crop residue cover has been given a specific meaning according to the table as below, which will be utilized in future studies.

| Crop residue cover | Cover |
| --- | --- |
| Conventional tillage | 0-0.15 |
| Low residue tillage | 0.15-0.30 |
| Conservation tillage | 0.30-0.60 |
| High residue tillage | 0.60-1.00 |

(Chesapeake Bay Program: Annapolis, MD, USA, 2016)

Thomason W, Duiker S, Ganoe K, et al. Conservation tillage practices for use in Phase 6.0 of the Chesapeake Bay Program watershed model[R]. CBP/TRS-308-16. Chesapeake Bay Program.< https://www. chesapeakebay. net/documents/CT_6. 0_Conservation_Tillage_ EP_Revised_Full_Report_12-14-16.2 _FINAL_NEW_TEMPLATE. pdf, 2016.

---

## Author Comment (AC5)

Dear Reviewer 4:

Thank you very much for your time involved in reviewing the manuscript and your very useful comments. This feedback greatly improved the quality of our paper and made this article more rigorous.

**SOM is closely related to the hot issue like climate change mitigation and sustainable agriculture development. In this study, the author aimed to mapped the spatial distributions of SOM in Northeast of China by random forest (RF) and evaluated the effects of the interaction of soil properties, land use and straw return on SOM spatial-temporal variation. Generally, the author reported a very interesting and valuable study. The manuscript is well organized and presented. The introduction provide sufficient background and the research design appropriate. The conclusions they reported can also well supported by the results. And I think the study is interest and valuable to the readers. However, there still some minor issues need deal with before the manuscript could be accepted for publication in Soil. Therefore I would like to suggest acceptance after minor modification. The detailed comment is as follow:**

**Line 26, you can only say SOM instead of SOM/SOC.**

Thank you for your comment. We have revised it.

**Line 52-53, please revise this sentence since it is confused. You can consider replace it with several simple sentence to make it more clear to the readers.**

Thank you for your comment. We have revised this sentence as follows:

"In the study, the overall objective was to take a typical black soil area as a case to quantify the relationship between SOM accumulation and straw return on a regional scale. This study area has a long-term straw return background."

**I would like suggest the author to explain the differences between GE and RF for quantifying the relative importance.**

Thank you for your comment. We have added this information in the section 3.3

**Line 105, I would like suggest the authors to provide more detailed and essential information of RF method.**

Thank you for the detailed comment. We have added more essential information about the RF method: "The RF model combines the predictions of all the individual trees, either by taking the majority vote in the case of classification or averaging the outputs in the case of regression. This approach helps to smooth out the noise in the data and produce more accurate predictions. It includes the number of trees (ntree) and the number of variables available for selection in each split (mtry)."

**The caption for the figures such as figure 4 need modification to make it more logical and grammar correct**

Thank you for your comment. We have focused the captions for figures and made some revisions.

**For the part of 3.5.1, it more like the Effects of soil types under the different the straw return on SOM variation**

Thank you for your comment. This section investigates the impact of straw return on soil organic matter (SOM) and its correlation with soil type. Various crop residue cover under different soil types were analyzed to determine their effects on SOM. The results confirm that the impact of straw return on SOM is indeed related to soil type, but this is largely due to the initial amount of SOM in the soil. Therefore, the study ultimately explores the effects of straw return on SOM variation under different soil types.

**Line 248-249, please modify this sentence**

Thank you for your time. We have revised it.

**Line 243, please revise this sentence**

Thank you for your time. We have revised it.

**I would like suggest the authors to introduce the meaning and value of their study more clearly in the end of Conclusion.**

Thank you for your comment. We have introduced the meaning and value of this study more clearly in the end of conclusion: "The study revealed that straw return is beneficial to carbon sink in farmland and is a better way to prevent a carbon source caused by the conservation of paddy field to dryland, which can contribute to the development of strategies to ensure the sustainability of agricultural soils."

**It should be better if the authors could provide more reasons for the ST variation of SOM in the survey region.**

Thank you for your comment. We have provided the reasons for the ST variation of SOM in the survey region in Section 2.1: "The study area was located in a black soil region. However, the soil in this region was threatened by land degradation. In view of this, a research base was established in Lishu County, Jilin Province, China, in 2007, and the straw return technology was popularized continually."